# Six-year (2016–2022) longitudinal patterns of mental health service utilization rates among children developmentally vulnerable in kindergarten and the COVID-19 pandemic disruption

**Fernanda Talarico**[1], **Dan Metes**[2], **Mengzhe Wang**[2], **Jake Hayward**[3], **Yang S. Liu**[1], **Julie Tian**[1], **Yanbo Zhang**[1], **Andrew J. Greenshaw**[1], **Ashley Gaskin**[4], **Magdalena Janus**[4], **Bo Cao**[1]*

**1** Department of Psychiatry, University of Alberta, Edmonton, Alberta, Canada, **2** Government of Alberta, Ministry of Health, Edmonton, Alberta, Canada, **3** Department of Emergency Medicine, University of Alberta Alberta, Canada, **4** Offord Centre for Child Studies, Department of Psychiatry and Behavioural Neurosciences, McMaster University, Canada

* cloudbocao@gmail.com

**Data Availability Statement:** Due to privacy policy restrictions individualized data cannot be shared.

## Abstract

### Introduction

In the context of the COVID-19 pandemic, it becomes important to comprehend service utilization patterns and evaluate disparities in mental health-related service access among children.

### Objective

This study uses administrative health records to investigate the association between early developmental vulnerability and healthcare utilization among children in Alberta, Canada from 2016 to 2022.

### Methods

Children who participated in the 2016 Early Development Instrument (EDI) assessment and were covered by public Alberta health insurance were included (N = 23 494). Linear regression models were employed to investigate the association between service utilization and vulnerability and biological sex. Separate models were used to assess vulnerability specific to each developmental domain and vulnerability across multiple domains. The service utilization was compared between pre- and post-pandemic onset periods.

### Results

The analysis reveals a significant decrease in all health services utilization from 2016 to 2019, followed by an increase until 2022. Vulnerable children had, on average, more events than non-vulnerable children. There was a consistent linear increase in mental health-

Data can be accessed with permission from both the Ministry of Health and the Ministry of Education in Alberta, Canada (https://www.alberta.ca/health-research).

**Funding:** This research was undertaken, in part, thanks to funding from the Canada Research Chairs program (BC), Alberta Innovates (BC and FT), Mental Health Foundation (BC), MITACS Accelerate program (BC and YL), Simon & Martina Sochatsky Fund for Mental Health (BC), Howard Berger Memorial Schizophrenia Research Fund (BC), the Abraham & Freda Berger Memorial Endowment Fund (BC), the Alberta Synergies in Alzheimer's and Related Disorders (SynAD) program (BC), University Hospital Foundation (BC) and University of Alberta (BC). The funders had no role in study design, data collection and analysis, decision to publish, or preparation of the manuscript.

**Competing interests:** The authors have declared that no competing interests exist.

related utilization from 2016 to 2022, with male children consistently experiencing higher utilization rates than females, particularly among vulnerable children. Specifically, there was a consistent linear increase in the utilization of anxiety-related services by children from 2016 to 2022, with females having, on average, 25 more events than males. The utilization of ADHD-related services showed different patterns for each group, with vulnerable male children having more utilization than their peers.

## Conclusion

Utilizing population-wide data, our study reveals sex specific developmental vulnerabilities and its impact on children's mental health service utilization during the COVID-19 pandemic, contributing to the existing literature. With data from kindergarten, we emphasize the need for early and targeted intervention strategies, especially for at-risk children, offering a path to reduce the burden of childhood mental health disorders.

### Author summary

The mental health of children and adolescents is increasingly recognized as a critical public health concern, especially in the wake of the COVID-19 pandemic. This study investigates mental health service utilization among children in Alberta, Canada, focusing on those identified as developmentally vulnerable during early childhood. Our research highlights a significant increase in mental health-related service use, particularly among developmentally vulnerable children, compared to their non-vulnerable peers. We also observed notable sex differences, with male children generally exhibiting higher utilization rates, although females showed a sharp increase in services post-pandemic. The findings emphasize the need for targeted interventions that address both developmental vulnerabilities and sex-specific needs. By linking early developmental assessments to later healthcare utilization, our study underscores the importance of early identification and support to mitigate long-term mental health challenges. These insights are crucial for policymakers and healthcare providers as they strive to enhance mental health services and ensure equitable access for all children, particularly those at heightened risk.

## Introduction

The escalating global prevalence of mental health disorders in children and adolescents has emerged as a pressing issue of our time [1]. Recent research has unveiled high rates of depression, anxiety, sleep disturbances, and posttraumatic stress symptoms, particularly in the aftermath of the COVID-19 pandemic [2–4]. The interplay of factors such as social isolation, disrupted routines, and economic uncertainties has exacerbated mental health problems, necessitating a re-evaluation of interventions spanning the developmental continuum from early childhood to adolescence [3,5,6].

The impact of early life adversities on mental health stands as a central concern warranting exploration. Pivotal experiences such as childhood maltreatment and socioeconomic hardships have been linked to an elevated risk of psychiatric conditions, including depression, anxiety, and substance use disorders [7–9]. Early life adversities not only have immediate mental

health outcomes but also contribute to persistent neurobiological alterations with lifelong implications [8–11].

Early intervention and prevention strategies become increasingly imperative in the achievement of alleviation of the burden of mental health disorders and promotion of optimal development [7,11,12]. While conventional approaches often target high-risk populations, emerging evidence highlights the potential of population-level interventions and early screenings to prevent the onset of more severe mental health challenges [13,14].

In this context, comprehending service utilization patterns and evaluating disparities in mental health-related service access among children holds paramount importance for informed policymaking and targeted interventions. This study aims to analyze the existing sex disparities in service utilization, especially those related to mental health, within the context of children's developmental characteristics. Specifically, we examine children who were recognized by kindergarten teachers as developmentally vulnerable, aiming to explore whether such vulnerabilities are associated with distinct service utilization patterns. We hypothesize that developmentally vulnerable children exhibit heightened levels of mental health-related service utilization compared to their non-vulnerable counterparts. Furthermore, considering the impact of the COVID-19 pandemic on healthcare utilization [2–4], the secondary hypothesis posits that the pandemic has led to an increase in mental health-related services utilization among both vulnerable and non-vulnerable children.

To comprehensively investigate the interplay between early developmental characteristics and mental health service utilization patterns, this study employs an established tool, the Early Development Instrument (EDI), measuring children's school readiness at school entry as reflected in their developmental status [15]. By integrating the EDI questionnaire data with healthcare utilization data, this study seeks to provide a nuanced understanding of whether children's developmental profiles predict patterns of mental health-related service utilization.

## Methods

### Study design

This was a 6-year longitudinal cohort study, linking teacher ratings of kindergarten children's development with subsequent administrative health records, specifically physician's office claims, emergency department visits, and hospitalizations.

The records are accessible through the databases available on the Alberta Ministry of Health website (www.alberta.ca/health-research.aspx).

This study was approved by the Research Ethics Boards, University of Alberta (Pro00072946).

### Participants

The study participants were drawn from the cohort of all 5–6 year-old children who participated in the 2016 Early Development Instrument (EDI) [15] assessment in Alberta, Canada and were covered by public Alberta Health insurance from 2016 to 2022 (N = 38,358). After applying exclusion criteria (more than 30% of missing data, under 30 days in the classroom, and missing parental consent) and data cleaning (no matching data with other administrative databases, children without Alberta biological records), 14,864 individuals were removed from the analysis. Thus, the eligible population for inclusion in the study comprises 23,494 (61.2% of the initial sample) children, which represents 21.4% of the age cohort in Alberta. More information on the inclusion and exclusion criteria is published elsewhere [16].

A prior analysis conducted by our research team highlighted significant differences between the final cohort of children (N = 23,494) and those excluded from the analysis, particularly

regarding socioeconomic status and mental health utilization [16]. Specifically, children excluded from the analysis due to non-participation in the EDI data collection or exclusion based on the defined criteria experienced a higher rate of subsidy (12.43%) in comparison to the cohort included in the final analysis (8.33%). Additionally, a slightly larger proportion of the excluded group demonstrated mental health utilization in physician claims (88.9%), compared to 88.2% among the final cohort of children. Importantly, no statistically significant differences were observed between these two groups concerning demographic characteristics and patterns of health service utilization.

The EDI assessment data collection in Alberta was conducted in February and March 2016. This instrument, widely recognized for its reliability and validity, offers an assessment of children's developmental vulnerabilities across multiple domains [17,18].

## Exposure

Teacher ratings on the Early Development Instrument were used to categorize the children into those who were developmentally vulnerable and those who were not. The EDI is a widely used and well-validated assessment tool in Canada that measures developmental vulnerability in children by identifying those whose skills and behaviours fall below the levels exhibited by most of their peers [19]. The assessment is completed by kindergarten teachers and consists of 103 items grouped into five developmental domains: physical health and well-being, social competence, emotional maturity, language and cognitive development, and communication and general knowledge [15]. The summary scores for each domain were calculated by averaging scores from domain-specific questions, with a range of 0 to 10, where higher scores indicate higher developmental status. A score falling on or below the 10th percentile of the distribution in a specific domain is considered "developmentally vulnerable," indicating a risk for difficulties. Children who scored in the "developmentally vulnerable" range in one or more domains are considered vulnerable overall [19]. Numerous studies showed that vulnerability on EDI is highly predictive of later academic achievement and social engagement in later grades [20–23] and mental health [20].

## Demographic variables

Vulnerable and non-vulnerable children were compared regarding their age, biological sex, and socioeconomic status, measured by whether the child was part of a subsidy group (i.e., whether the child was part of a subsidy group in 2015/16).

## Outcome

The primary variable of interest in this study is the number of all conditions and mental health-related services utilized during the study period (2016 to 2022). We used diagnosis codes from the International Classification of Diseases, 9th and 10th revisions (ICD-9 [24] and ICD-10 [25]) available in service utilization records to identify mental health disorders and their sub-conditions. We selected the top three mental health disorders in reference to the highest number of children seeking treatment (namely anxiety disorders, mood disorders, and ADHD). A complete list of the diagnosis codes used is available in S1 Table.

To account for changes in population size over time in Alberta, we calculated crude rates per 1,000 population. Specifically, we divided the total number of all events and mental health-related events by the corresponding population size in each group (i.e., vulnerability and biological sex groups). We then multiplied the resulting value by 1,000 to obtain the crude rate per 1,000 population for each group in each year.

### Analysis

Linear regression models with vulnerability group (yes or no), biological sex (male or female), and year as predictor variables were used to investigate the association between the number of all events and mental health-related events (i.e., dependent variables) with time using. The interaction term between vulnerability and sex was also included as an independent variable. Separate models were conducted to assess vulnerability specific to each developmental domain, as well as vulnerability across multiple domains (i.e., vulnerability in one or more domains). We adjusted the p-values using false discovery rate (FDR) and models with adjusted p-values lower than 0.05 were considered statistically significant.

Additionally, we compared service utilization between the pre- and post-pandemic onset periods. The mean value of service utilization from January 2016 to February 2020 was computed and defined as the 'pre-pandemic' period, while the mean value from March 2020 until December 2022 was classified as the 'post-pandemic onset' period. To analyze the variability within each group, the standard deviation was calculated for each time point and compared between the pre- and post-onset periods.

The statistical analyses and graphical representations were performed using R version 4.1.1 [26] in July 2023.

## Results

The mean age of vulnerable and non-vulnerable children was 5.63 and 5.70, respectively. Among the 23,494 children, 11,289 (48.1%) were females and 6,702 were classified as vulnerable in one or more development domains. Among vulnerable children, 9.75% of all health service utilization were related to a mental health problem (N = 3,719) prior to 2022. This number rose to 24.2% in 2022 (N = 7,050).

There is a statistically significant difference in biological sex distribution between the two groups (Table 1). Despite there being more females in the overall cohort, males represent the majority among the vulnerable group (Table 1).

The percentage of mental health utilization increased over time for all groups, with vulnerable children exhibiting a consistently higher rate than their non-vulnerable peers across all years (Table 2).

### All conditions

The results indicate a significant linear decrease in all health services utilization from 2016 to 2020, followed by a significant reduction in 2020. Utilization slowly increased thereafter, reaching levels similar to those observed pre-pandemic by 2022 (Fig 1A). Vulnerable children, on average, had 648 more events than non-vulnerable children ($\beta_{vulnerability}$ = 647.9; p-

**Table 1. Cohort characteristics.**

| Variable Name | Variable Label | Non-Vulnerable Children (n = 16,792)—n (%) | Vulnerable Children (n = 6,702) -n (%) | p-value# |
|---|---|---|---|---|
| Age–mean (SD) | Years | 5.70 (0.32) | 5.63 (0.34) | <0.001 |
| Child's Biological Sex | Female | 8869 (52.8%) | 2420 (36.1%) | <0.001 |
| | Male | 7923 (47.2%) | 4282 (63.9%) | |
| Socioeconomic/ Subsidy Status (Child) | Subsidy | 900 (5.4%) | 1042 (15.7%) | <0.001 |
| | No Subsidy | 15,800 (94.6%) | 5597 (84.3%) | |

SD = standard deviation; n = sample size.

**Table 2. Descriptive characteristics of service utilization by sex and year.**

| Year | Vulnerability in one or more domains | MH utilization—n (%) | All utilization—n |
|------|--------------------------------------|----------------------|-------------------|
| 2016 | Y | 3713 (9.75%) | 38068 |
|      | N | 1792 (2.44%) | 73451 |
| 2017 | Y | 5130 (14.2%) | 36057 |
|      | N | 5130 (4.61%) | 69470 |
| 2018 | Y | 6021 (17.7%) | 34060 |
|      | N | 4123 (6.31%) | 65344 |
| 2019 | Y | 6526 (19.8%) | 32989 |
|      | N | 5354 (8.10%) | 66122 |
| 2020 | Y | 5986 (23.6%) | 25413 |
|      | N | 5863 (11.3%) | 51778 |
| 2021 | Y | 6634 (24.5%) | 27113 |
|      | N | 7365 (12.9%) | 56964 |
| 2022 | Y | 7050 (24.2%) | 29145 |
|      | N | 8186 (13.5%) | 60787 |

Y = yes (vulnerable children); N = no (non-vulnerable children); n = sample size or number of events (utilization).

value = 0.013). Almost all utilization involved office visits (Fig 1A and 1B) and therefore the patterns of association for each of these two variables are similar: there is a linear decrease in utilization ($\beta_{time}$ = -85.7; p-value = 0.018) and vulnerable children, on average, had 670 more events than non-vulnerable children ($\beta_{vulnerability}$ = 569.9; p-value = 0.010).

Similar to office visits, emergency visits exhibited a linear decrease ($\beta_{time}$ = -28.7; p-value = 0.003) from 2016 to 2020, and a gradual increase thereafter (Fig 1C). The hospitalization patterns of vulnerable children were similar to those observed in emergency visits, with a linear decrease from 2016 to 2020. However, in 2022, there was a sharp increase in hospitalizations among females, surpassing the number of hospitalizations in males, as illustrated in Fig 1D. Non-vulnerable children, on the other hand, exhibited a slight decrease in hospitalizations from 2016 to 2021 and in 2022, while the hospitalization rate of males continued to decrease, that of females increased, exceeding the rate of males. Vulnerable children had, on average, eight more hospitalization events than their non-vulnerable peers, which was statistically significant ($\beta_{vulnerability}$ = 8.1; p-value = 0.008).

Consistent patterns of association were observed across distinct developmental domains. Generally, children classified as vulnerable in areas such as communication and general knowledge (CG), emotional maturity (EM), language and cognitive development (LC), physical health and well-being (PH), and social competence (SOC) had a higher number of office visits compared to non-vulnerable children. Children with vulnerabilities in EM, LC, PH, and SOC domains had higher rates of emergency department visits and hospitalizations. A comprehensive breakdown of the findings, indicating the domain-specific beta and p-values for office visits, emergency department visits, and hospitalizations, is shown in S2 Table.

In terms of the average number of events that occurred during the pre-pandemic period (2016–2019) and post-onset period (2020–2022), both vulnerable and non-vulnerable children experienced a decline in healthcare services utilization after the onset of the COVID-19 pandemic, as shown in S1 Fig. During the post-onset period, there were no substantial differences in utilization between the two groups, except for hospitalizations, in which vulnerable children continued to have higher rates than non-vulnerable children. In terms of biological sex

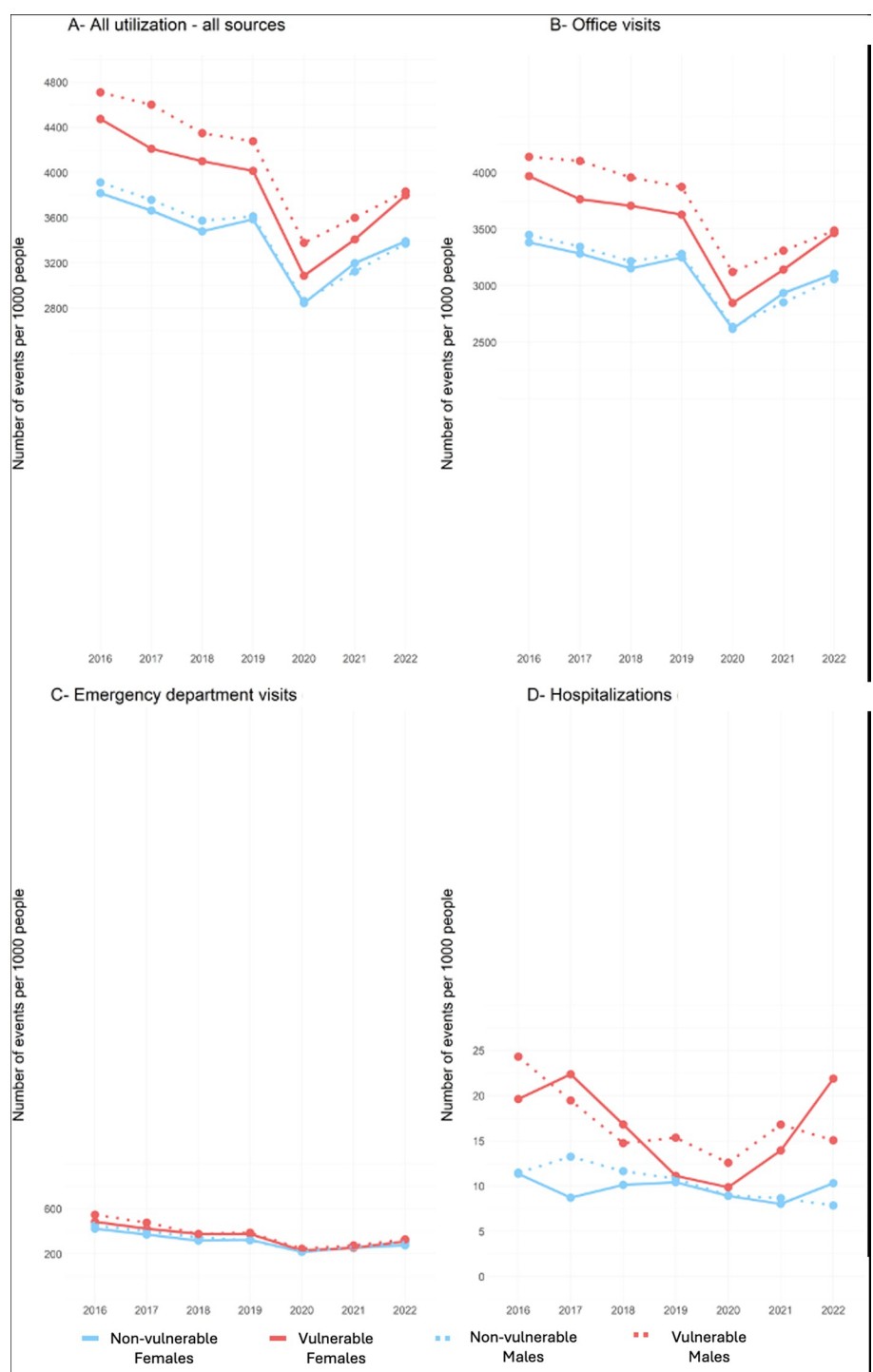

**Fig 1. Trend of all service utilization for all conditions between 2016 and 2022.** Note that D is not on the same scale as A, B, and C.

differences, male children had slightly higher utilization rates than females for all utilization sources and emergency department visits in both periods. However, females showed slightly higher utilization rates than males for office visits and hospitalizations in the post-onset period.

## Mental health conditions

Our findings demonstrate a consistent linear increase in the utilization of all mental health-related services between 2016 and 2022. Throughout the years, male children consistently displayed higher utilization rates than females, particularly among vulnerable children (Fig 2A). Overall, the results of the linear regression analysis indicated that, on average, vulnerable male

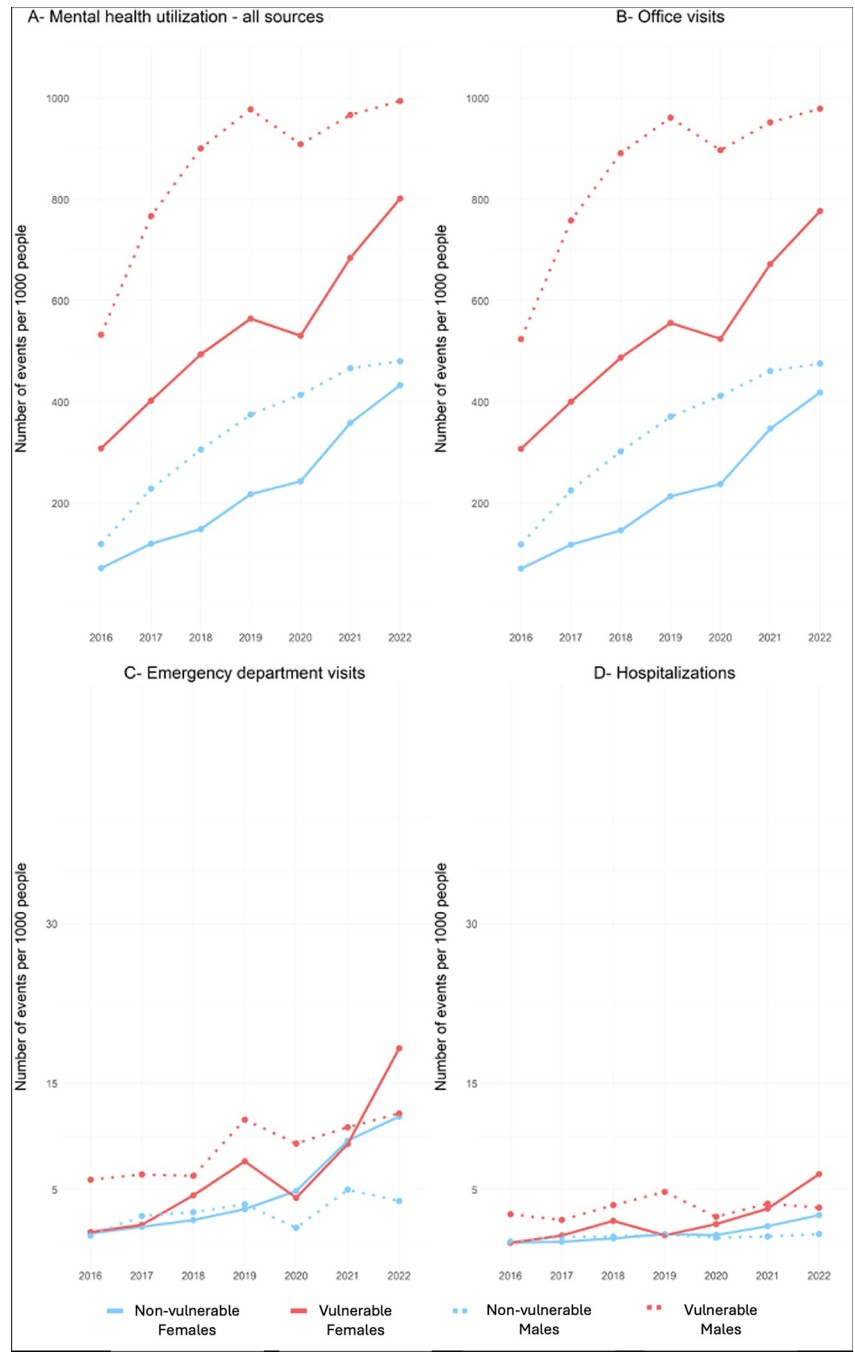

**Fig 2. Trend of mental health-related services utilization between 2016 and 2022.** Note that A and B are in different scales than C and D.

children had significantly more events than non-vulnerable males ($\beta_{vulnerability*sex(M)}$ = 209.4; p-value = 0.002).

The data indicate that office visits constitute the majority of mental health-related utilization (Fig 2B) and the findings are consistent with the overall mental health utilization results. Specifically, the analysis revealed that vulnerable male children had more events than non-vulnerable males, which was statistically significant ($\beta_{vulnerability*sex(M)}$ = 203.7; p-value = 0.002).

There was also a consistent increase in mental health-related emergency visits between 2016 and 2022. Although there was a slight decline in 2020 for vulnerable male children, their emergency visits resumed a steady increase until 2022 (Fig 2C). In contrast, vulnerable female children showed a significant increase in emergency visits in the post-pandemic-onset period, surpassing the values observed for vulnerable males. There were relatively stable emergency visit rates among non-vulnerable males throughout the years, with a slight decrease in mental health-related visits in 2020 and a small increase in 2021. In contrast, among female non-vulnerable children there was a consistent increase in emergency visits from 2016 to 2020, with a sharp increase in subsequent years. In 2020 and 2021, their numbers were comparable to those of vulnerable female children, and in 2022, they were comparable to those of vulnerable male children, as demonstrated in Fig 2C.

The linear regression analysis revealed that the increasing trend of mental health-related emergency visits was significant for vulnerable male children compared to their peers ($\beta_{vulnerability*sex(M)}$ = 4.3; p-value = 0.035). Although the mental health-related hospitalization pattern was similar to those observed in emergency visits, it was not statistically significant.

Different outcomes were observed across various developmental domains. Children identified as vulnerable in the language and cognitive development (LC) and social competence (SOC) domains exhibited elevated frequencies of office visits, emergency department visits, and hospitalizations. Vulnerability in the emotional maturity (EM) and physical health (PH) domains was associated with higher numbers of office visits. Additionally, vulnerability in the communication and general knowledge (CG) domain was linked to increased office visits and emergency department visits. Detailed results of this analysis are in S3 Table.

To better understand the impact of the COVID-19 pandemic on mental health utilization rates, we analyzed the average number of events that occurred during the pre-pandemic period (2016–2019) and post-onset period (2020–2022), along with their corresponding standard errors (SEs). Our results revealed that among both vulnerable and non-vulnerable children, there was an increase in healthcare services utilization (Fig 3A–3D). Specifically, for all sources and office visits, both groups and sexes demonstrated a similar increase, with vulnerable male children consistently exhibiting the highest utilization rates. However, there were notable sex differences in the rates of emergency department visits and hospitalizations. While male utilization remained relatively stable between the pre- and post-onset periods, female utilization increased significantly. Both vulnerable and non-vulnerable female children displayed a similar increase in emergency department visits, but the hospitalizations among vulnerable female children increased at a higher rate compared to female non-vulnerable children.

## Specific mental health conditions

There is a consistent linear increase in the utilization of anxiety-related services by vulnerable and non-vulnerable children from 2016 to 2022, as shown in Fig 4A. A similar trend was observed among both male and female children between 2016 and 2020. However, the slope for females increased in 2021 and they surpassed males in utilization rates. Our linear regression analysis indicated that males had fewer anxiety-related events than females over the years

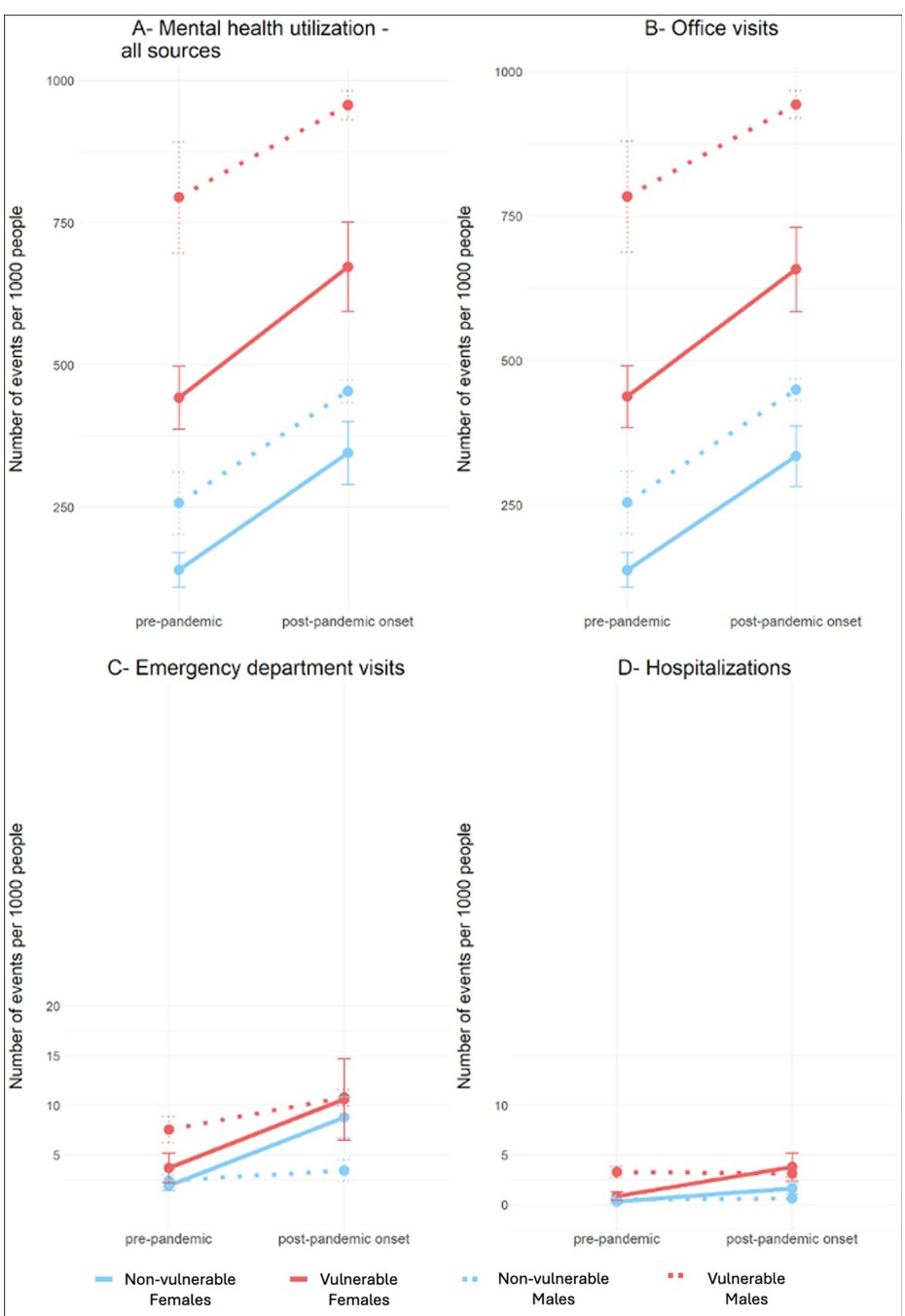

**Fig 3. Comparison of mental health-related services utilization between the pre (2016–2019) and post (2020–2022) pandemic onset periods.** Note that A and B are on different scales than C and D. The error bar indicates standard error.

($\beta_{sex(M)}$ = -25.2; p-value = 0.032). Additionally, vulnerable children had more events than their peers ($\beta_{vulnerability*time}$ = 9.0; p-value = 0.032).

There was a consistent and gradual rise in mood disorders-related utilization from 2016 to 2020, which was followed by a significant increase in subsequent years. This increase was particularly pronounced among vulnerable female children (Fig 4B), but there were no statistically significant differences observed between groups.

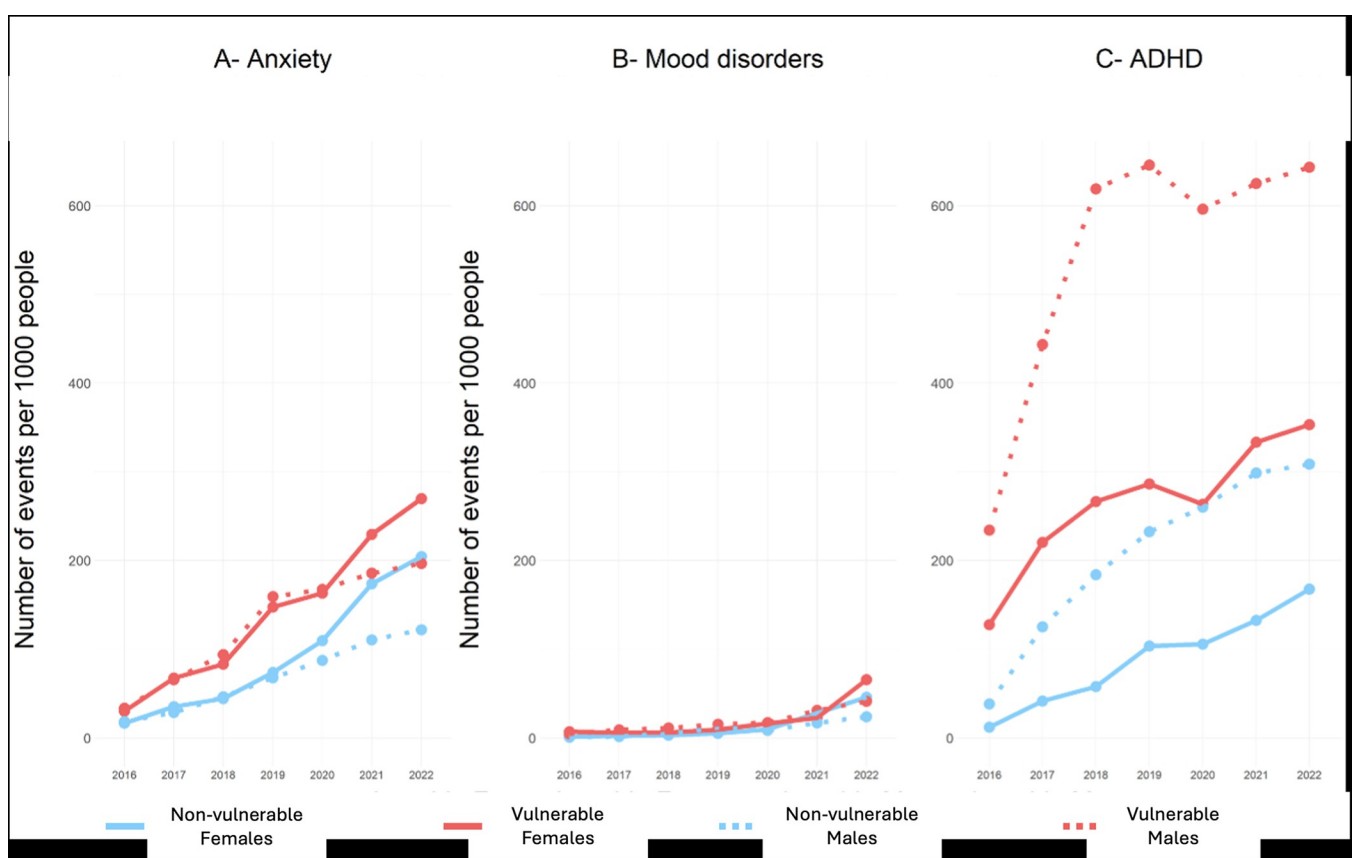

**Fig 4. Trend of specific mental health disorders for all services utilization between 2016 and 2022.**

The utilization of ADHD-related services showed different patterns for each group. There was a stable increase for females between 2016 and 2022, while vulnerable males had a sharp increase between 2016 and 2018 and then stabilized. Non-vulnerable males showed a steady linear increase throughout the years, similar to vulnerable females' pattern (Fig 4C). Our analysis further revealed that vulnerable male children had more events related to ADHD than their peers ($\beta_{\text{vulnerability*sex(M)}}$ = 161.5; p-value = 0.002).

The analysis based on specific developmental domains revealed an association between vulnerability and mental health conditions (i.e., ADHD, mood disorder, and anxiety) in all domains, indicating a higher utilization rate among vulnerable children as well as male children. Among vulnerable children, only the physical health (PH) domain exhibited a significant association with increased utilization. For more detailed information, please refer to S4 Table.

Our analysis of healthcare utilization rates before and after the onset of the COVID-19 pandemic found that utilization rates increased after the pandemic onset for all children in all three mental health conditions (Fig 5A–5C). The rate of increase was similar across all groups, indicating that the pandemic did not affect the service use patterns of vulnerable and non-vulnerable children differentially.

## Discussion

Our study explored the utilization patterns of mental health-related services among 6–12-year-old children in Alberta with known kindergarten teacher-rated developmental vulnerability

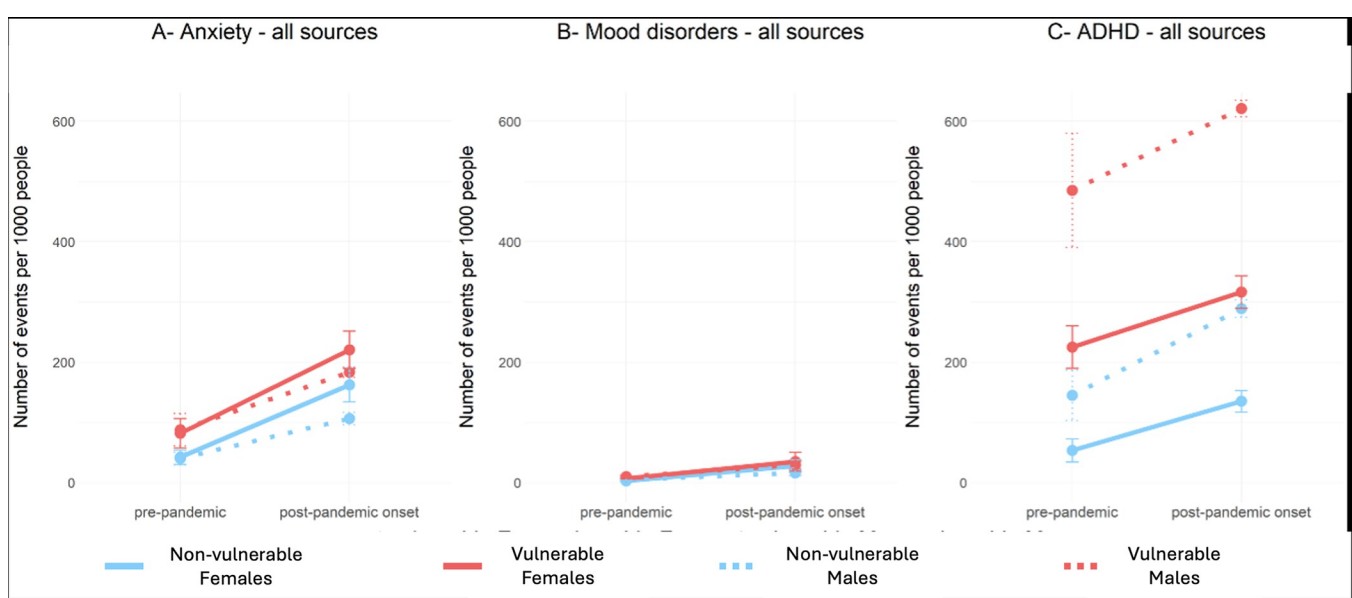

**Fig 5. Comparison of mental health-related service utilization between the pre (2016–2019) and post (2020–2022) pandemic onset periods.** The error bar indicates standard error.

status prior to and after the onset of the COVID-19 pandemic. Our findings suggest that the COVID-19 pandemic had a similar impact on healthcare utilization rates for both vulnerability groups, except for hospitalizations. We observed that developmentally vulnerable children demonstrated higher engagement with mental health-related services compared to their non-vulnerable counterparts across the whole period studied. These findings underline the demand for targeted interventions and comprehensive support strategies for vulnerable populations, underscoring the imperative of equitable access to mental health services. This includes measures to bridge potential sex and vulnerability disparities in access, such as fortifying community-based support networks and enhancing mental health programs within educational institutions. Our study highlights the importance of ensuring that vulnerable populations have the same opportunities to benefit from these services as their non-vulnerable counterparts.

Furthermore, we demonstrated a COVID-19 pandemic-related disruption in utilization patterns among both vulnerable and non-vulnerable children, which is in line with the evidence of the global surge in mental health disorders among children and adolescents, including depression, anxiety, sleep disorders, and posttraumatic stress symptoms, reflects the pandemic's consequences [1–4]. Our study not only adds to this evidence but also reveals how early developmental vulnerabilities can interact with such a disruption, thereby creating distinct challenges in accessing mental health services.

The role of age emerges as a crucial factor in comprehending the escalating patterns of mental health issues observed among children. As children progress through developmental stages, specific disorders become more evident and diagnosable. Moreover, advancing age exposes them to an array of stressors and complexities, potentially contributing to the emergence of additional health concerns, which in turn may explain the increasing patterns of utilization observed in our study. This perspective aligns with existing research that accentuates the role of age in influencing mental health outcomes [27,28].

Our findings also align with established literature that depicts a consistent upward trend in mental health challenges among children [1,4]. The high prevalence of depression, anxiety, and other mental health issues in adolescents during the pandemic is well-documented [3,4,6].

Our study's unique contribution to the understanding of the pandemic's association with children and adolescent mental health is the ability to distinguish between groups of children who were rated as vulnerable in their developmental health, broadly reflecting school readiness at school entry. While patterns showed some variation in relation to type of service, in general, children who were developmentally vulnerable at 5 years of age, experienced higher engagement with health services prior to, but especially after the pandemic onset, than those who were not vulnerable. By identifying sex disparities in utilization, our study supplies insights for policymakers and healthcare providers in addressing the unique service needs of these vulnerable populations.

Our study used a large population-wide dataset on health service utilization, with a unique link to data on children's early vulnerability. Nevertheless, it has some important limitations. First, no detailed family socioeconomic data were available, which can influence patterns of health service utilization, especially in vulnerable populations. Poor socioeconomic status, represented as low income and poorer education, is known to be associated with reduced healthcare access and utilization. Second, we were unable to distinguish between rural and urban areas in our analysis. This is a significant limitation because healthcare access and service availability can vary considerably between the two settings. Also, we excluded children who did not participate in the EDI collection and based on specific criteria. As noted in our previous study [16], these children presented different patterns of service utilization. Finally, while the linkage of EDI data with health service utilization revealed important associations, it limited the sample size, affecting the generalizability of our findings.

## Conclusion

Our study found sex specific developmental vulnerabilities and generally higher health system utilizations related to mental health in children with developmental vulnerabilities at the population level during the COVID-19 pandemic. Based on our findings, we suggest that tailored interventions aimed at vulnerable populations are essential. This may encompass strategies such as broadening telehealth options, fortifying community-based support networks, and enhancing mental health programs within educational institutions. Future studies should examine the impact of these strategies on children's outcomes. Due to its data linkage from kindergarten, our study highlights the potential opportunities for early intervention and prevention strategies, especially for children at an elevated risk. By prioritizing these recommendations, policymakers can take substantial steps toward reducing the burden of mental health disorders among children.

## Disclaimer

This study is based in part on data provided by Alberta Health. The interpretation and conclusions contained herein are those of the researchers and do not necessarily represent the views of the Government of Alberta. Neither the Government nor Alberta Health expresses any opinion about this study.

## Supporting information

**S1 Table. List of ICD 9 and 10 codes for mental health conditions.**
(DOCX)

**S2 Table. Results of linear regression models for domain-specific analysis of all utilization.**
(DOCX)

**S3 Table. Results of linear regression models for domain-specific analysis of mental health-related utilization.**
(DOCX)

**S4 Table. Results of linear regression models for domain-specific analysis of specific mental health disorders.**
(DOCX)

**S1 Fig. Comparison of health services utilization between the pre (2016–2019) and post (2020–2022) pandemic onset periods.**
(DOCX)

## Author Contributions

**Data curation:** Dan Metes, Mengzhe Wang.

**Formal analysis:** Fernanda Talarico.

**Funding acquisition:** Bo Cao.

**Methodology:** Fernanda Talarico, Magdalena Janus.

**Project administration:** Mengzhe Wang, Andrew J. Greenshaw.

**Resources:** Andrew J. Greenshaw.

**Supervision:** Yanbo Zhang, Andrew J. Greenshaw, Bo Cao.

**Validation:** Ashley Gaskin, Magdalena Janus.

**Visualization:** Fernanda Talarico.

**Writing – original draft:** Fernanda Talarico.

**Writing – review & editing:** Fernanda Talarico, Dan Metes, Mengzhe Wang, Jake Hayward, Yang S. Liu, Julie Tian, Ashley Gaskin, Magdalena Janus, Bo Cao.

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
