## [Decision Letter · Decision Letter 0]

20 May 2024

PDIG-D-23-00471

Six-year (2016-2022) longitudinal patterns of mental health service utilization rates among children developmentally vulnerable in kindergarten and the COVID-19 pandemic disruption

PLOS Digital Health

Dear Dr. Cao,

Thank you for submitting your manuscript to PLOS Digital Health. After careful consideration, we feel that it has merit but does not fully meet PLOS Digital Health's publication criteria as it currently stands. Therefore, we invite you to submit a revised version of the manuscript that addresses the points raised during the review process.

Please submit your revised manuscript within 30 days Jun 19 2024 11:59PM. If you will need more time than this to complete your revisions, please reply to this message or contact the journal office at digitalhealth@plos.org. Please include the following items when submitting your revised manuscript:

We look forward to receiving your revised manuscript.

Kind regards,

Margaret Isioma Ojeahere, MBBS, FWACP

Academic Editor

PLOS Digital Health

Journal Requirements:

2. Please send a completed 'Competing Interests' statement, including any COIs declared by your co-authors. If you have no competing interests to declare, please state "The authors have declared that no competing interests exist". Otherwise please declare all competing interests beginning with the statement "I have read the journal's policy and the authors of this manuscript have the following competing interests:"

3. Please amend your detailed Financial Disclosure statement. This is published with the article. It must therefore be completed in full sentences and contain the exact wording you wish to be published.

4. Please provide separate figure files in .tif or .eps format only and remove any figures embedded in your manuscript file. Please also ensure that all files are under our size limit of 10MB.

Additional Editor Comments (if provided):

Reviewers' comments:

Reviewer's Responses to Questions

**Comments to the Author**

1. Does this manuscript meet PLOS Digital Health’s publication criteria? Is the manuscript technically sound, and do the data support the conclusions? The manuscript must describe methodologically and ethically rigorous research with conclusions that are appropriately drawn based on the data presented.

Reviewer #1: Yes

Reviewer #2: Yes

Reviewer #3: Yes

2. Has the statistical analysis been performed appropriately and rigorously?

Reviewer #1: Yes

Reviewer #2: Yes

Reviewer #3: Yes

3. Have the authors made all data underlying the findings in their manuscript fully available (please refer to the Data Availability Statement at the start of the manuscript PDF file)?

Reviewer #1: No

Reviewer #2: No

Reviewer #3: Yes

4. Is the manuscript presented in an intelligible fashion and written in standard English?

Reviewer #1: Yes

Reviewer #2: Yes

Reviewer #3: Yes

5. Review Comments to the Author

Reviewer #1: important topic and good contribution from the research to help address and design interventions / preventions in more targeted population taking in consideration gender and pre vulnerability

Reviewer #2: A. General comment:

This is a good study which I recommend for publication following the correction of minor errors highlighted below;

B. Specific comments

Query 1: under methods subsection participants

 Line 104, the authors need to recheck the percentage quoted as eligible population included in the study (61.1%). This should be 23496/38358 = 61.2% from the figure presented.

Query 2: under methods subsection participants 

 Line 106: The authors need to be consistent with a referencing style. Hence, replace the Talarico eta al 2023 with Vancouver referencing (numeric referencing style).

Query 3: under Result section

Line 171-172: ‘Among the 23494 children, there were 2445 (29.7%) females’. This frequency and percentage quoted need to be rechecked by the authors. Overall (i.e vulnerable and non-vulnerable), Females constituted 48% (11289/23494) as presented in Table 1. Likewise, the mean age should be rechecked . Kindly recheck these and make necessary adjustment or provide clarity to the frequency and percentages earlier quoted above.

Query 4: under Result section

 Line 186, Table 2: The Column Letter ‘N’ and the Row ‘N’ need to be delineated and well defined. The authors defined the ‘N’ and ‘Y’ on the row at the bottom of the table but this is not so for that on the Column. I suggest the authors use smaller ‘n’ for that of the column and define it at the bottom of the table for clarity.

Reviewer #3: The paper highlighted the importance of the escalating global prevalence of mental health disorders in children and adolescents as well as the impact of early life adversities on mental health. 

Specifically, the study explored the utilization patterns of mental health-related services among 6–12-year-old children in Alberta, with known kindergarten teacher-rated developmental vulnerability status, prior to and after the onset of the COVID-19 pandemic.

The findings showed that developmentally vulnerable children demonstrated higher engagement with mental health-related services compared to their non vulnerable counterparts across the whole time period studied, underlining the demand for targeted interventions and comprehensive support strategies for vulnerable populations, and underscoring the imperative of equitable access to mental health services. 

The paper is therefore relevant and contributes to the body of knowledge. 

The paper generally followed the structure of a scientific article. 

The introduction was appropriate and adequately provided enough background knowledge about the subject matter. 

The authors spent talked about the children excluded from the study and even made some comparison with those included in the study. This should not be necessary since those children were excluded from this study. If the authors want to write another paper on such comparison, it will be a good idea. 

There was wrong referencing style while talking about the publication on the inclusion and exclusion criteria. i.e (Talarico et al., 2023).

Some of the figures and percentages in the results were difficult to understand. The figure, 2,445 is reported as 29.7% females. However, 2445 is not 29.7% of 23,494. The authors should explain and clarify the figures with the percentage.

The figures should follow the narratives so that it can be easy to relate what is read to the figure for easy understanding.

The tables are not clear enough to understand. The table needs to be better presented with the columns and rows clearly delineated to clearly understand the information presented even by a layman. 

Overall the study is a good one and has contributed to the body of knowledge. 

Overall the study is a good one and has contributed to the body of knowledge

6. PLOS authors have the option to publish the peer review history of their article (what does this mean?). If published, this will include your full peer review and any attached files.

**Do you want your identity to be public for this peer review?** For information about this choice, including consent withdrawal, please see our Privacy Policy.

Reviewer #1: Yes: Dr Yacine HADJIAT

Reviewer #2: No

Reviewer #3: Yes: Kingsley Mayowa Okonoda

---

## [Decision Letter · Decision Letter 1]

12 Aug 2024

Six-year (2016-2022) longitudinal patterns of mental health service utilization rates among children developmentally vulnerable in kindergarten and the COVID-19 pandemic disruption

PDIG-D-23-00471R1

Dear Dr Cao,

We are pleased to inform you that your manuscript 'Six-year (2016-2022) longitudinal patterns of mental health service utilization rates among children developmentally vulnerable in kindergarten and the COVID-19 pandemic disruption' has been provisionally accepted for publication in PLOS Digital Health.

Best regards,

Margaret Isioma Ojeahere, MBBS, FWACP

Academic Editor

PLOS Digital Health

Reviewer Comments (if any, and for reference):

Reviewer's Responses to Questions

**Comments to the Author**

1. If the authors have adequately addressed your comments raised in a previous round of review and you feel that this manuscript is now acceptable for publication, you may indicate that here to bypass the “Comments to the Author” section, enter your conflict of interest statement in the “Confidential to Editor” section, and submit your "Accept" recommendation.

Reviewer #2: All comments have been addressed

Reviewer #3: All comments have been addressed

2. Does this manuscript meet PLOS Digital Health’s publication criteria? Is the manuscript technically sound, and do the data support the conclusions? The manuscript must describe methodologically and ethically rigorous research with conclusions that are appropriately drawn based on the data presented.

Reviewer #2: Yes

Reviewer #3: Yes

3. Has the statistical analysis been performed appropriately and rigorously?

Reviewer #2: Yes

Reviewer #3: Yes

4. Have the authors made all data underlying the findings in their manuscript fully available (please refer to the Data Availability Statement at the start of the manuscript PDF file)?

Reviewer #2: Yes

Reviewer #3: Yes

5. Is the manuscript presented in an intelligible fashion and written in standard English?

Reviewer #2: Yes

Reviewer #3: Yes

6. Review Comments to the Author

Reviewer #2: The authors have addressed all the queries raised in the previous review and this has further improves the quality of the manuscript. I recommend it for publication in the journal.

Reviewer #3: (No Response)

7. PLOS authors have the option to publish the peer review history of their article (what does this mean?). If published, this will include your full peer review and any attached files.

**Do you want your identity to be public for this peer review?** For information about this choice, including consent withdrawal, please see our Privacy Policy.

Reviewer #2: **Yes: **Dr Mumeen Olaitan SALIHU

Reviewer #3: **Yes: **Kingsley Mayowa Okonoda
